# Septic Pulmonary Emboli Detected by ^18^F-FDG PET/CT in a Patient with Central Venous Catheter-Related *Staphylococcus aureus* Bacteremia

**DOI:** 10.3390/diagnostics12102479

**Published:** 2022-10-13

**Authors:** Jang Yoo, Miju Cheon

**Affiliations:** Department of Nuclear Medicine, VHS Medical Center, Seoul 05368, Korea

**Keywords:** septic pulmonary emboli, catheter-related bloodstream infection, ^18^F-FDG PET

## Abstract

We describe a case of ^18^F-FDG PET/CT detecting septic pulmonary emboli in a patient with *Staphylococcus aureus* catheter-related bloodstream infection (CRBSI). The patient, who had an implantable venous access port for chemotherapy, underwent ^18^F-FDG PET/CT to diagnose unsuspected infectious foci. The PET/CT examination made it possible to offer a suggestive diagnosis and yielded metastatic infectious foci.

**Figure 1 diagnostics-12-02479-f001:**
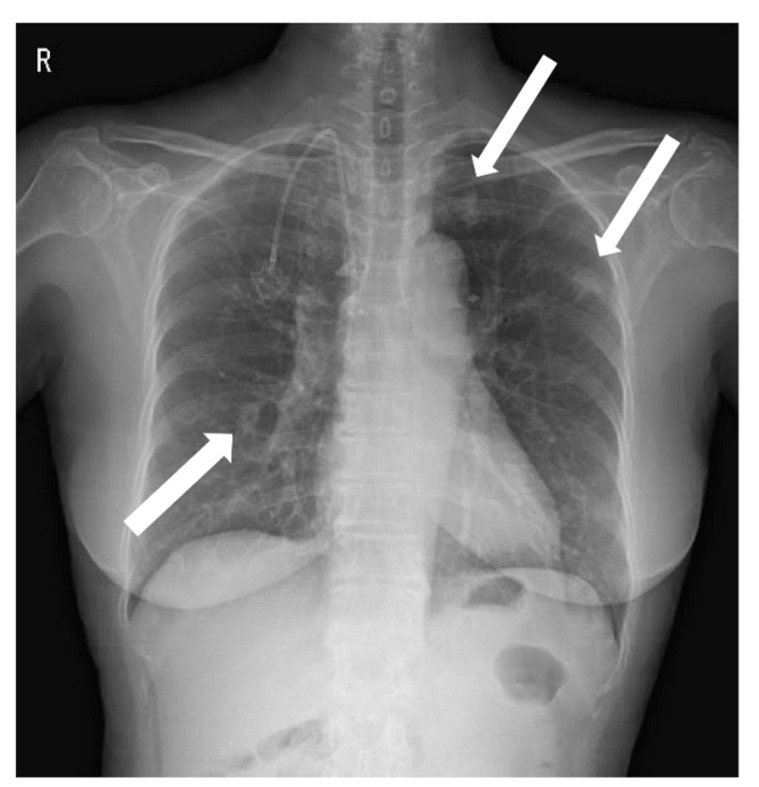
A chest X-ray was performed at admission, showing scattered nodular opacities in the left lung upper lobe and right lung lower lobe, which indicated a central area of excavation (arrows). ^18^F-fluorodeoxyglucose positron emission tomography/computed tomography (^18^F-FDG PET/CT) has increasingly been used to manage cancers and infections [1,2,3]. Since FDG uptake is directly representative of glucose metabolism, it can increase in inflammatory cells as well as tumor cells. Long-term indwelling central venous catheters are necessary for treating cancer patients due to chemotherapy. They depend on their central venous catheters daily, which could predispose a significant risk of complications such as catheter-related bloodstream infection (CRBSI) [4,5,6]. CRBSI can be complicated by metastatic infectious foci associated with a high morbidity and mortality rate, which should require prolonged systemic antimicrobial treatment [7]. The significant complication of CRBSI is septic thrombosis, with a prevalence of 15~24% [8,9]. The clinical diagnosis of septic foci is critical but may be difficult to establish due to the challenge of determining between sterile catheter-related thrombosis and actual septic thrombosis. Additionally, symptoms are often non-specific, and there is a lack of sensitivity to conventional diagnostic imaging techniques. Only a few studies investigated that ^18^F-FDG PET/CT can find the infectious foci, demonstrating it as an accurate imaging modality for metastatic foci [10,11,12]. Here, we would like to report a female patient with a *Staphylococcus aureus*-implantable venous access catheter infection in which ^18^F-FDG PET/CT determined unsuspected septic pulmonary emboli. A 71-year-old female patient with known ovarian cancer visited our hospital to receive the 4th adjuvant chemotherapy. She was treated with total abdominal hysterectomy, bilateral salpingo-oophrectomy and omentectomy 6 months ago. She also had a history of central venous catheterization by the right internal jugular vein approach, terminating at the junction of the superior vena cava and right atrium. The adjuvant chemotherapy was already performed three times as the regimen of Paclitaxel plus Carboplatine after surgery. At admission, she presented no clinical symptoms such as fever, cough, sputum, dyspnea, or chest pain. However, the chest X-ray showed scattered nodular opacities in the left lung upper lobe and right lung lower lobe, suspicious of metastatic nodules (Figure 1). Her blood test showed unexplained leukocytosis (12.58 × 10^3^/μL), elevated D-dimer (9.46 mg/L), and a tumor marker such as CA-125 (41.89 U/mL). ^18^F-FDG PET/CT was performed 2 days after admission, observing abnormal FDG uptake in the chemo-port catheter, right pectoralis muscle, and 1st costochondral junction (Figure 2a–d). PET/CT also revealed hypermetabolic nodules scattered throughout both lungs, consistent with septic embolism (Figure 2e). In the evening of the day of the PET/CT examination, the patient presented swelling, redness, and some discomfort at the catheter insertion site. The catheter was removed the next day, isolating methicillin-sensitive *Staphylococcus aureus* from the catheter tip and peripheral vein. A transthoracic echocardiogram and fundus examination, which were performed to evaluate possible metastatic infections such as infective endocarditis or endophthalmitis, were reported as normal. She was treated with intravenous cefazolin for 4 weeks followed by oral linezolid for 1 week, recovering uneventfully without relapse. She was also treated with rivaroxaban 15 mg for 3 weeks and was diagnosed with pulmonary thromboembolism. After 2 months, a follow-up chest CT showed that the pre-existing multiple nodules in both lungs had disappeared (Figure 3).

**Figure 2 diagnostics-12-02479-f002:**
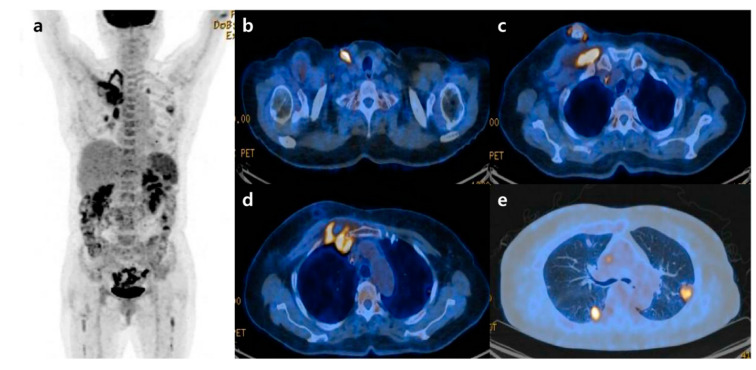
^18^F-FDG PET/CT images (**a**) showed abnormal FDG uptake in the chemo-port catheter (**b**), right pectoralis muscle (**c**), and the anterior thoracic wall around the first costochondral junction, consistent with an inflammatory/infectious process (**d**). Lung window setting image, showing multiple cavitating hypermetabolic nodules in both lungs consistent with septic pulmonary emboli (**e**). According to these findings, the patient was suspected of having catheter-related bloodstream infection and septic pulmonary emboli.

**Figure 3 diagnostics-12-02479-f003:**
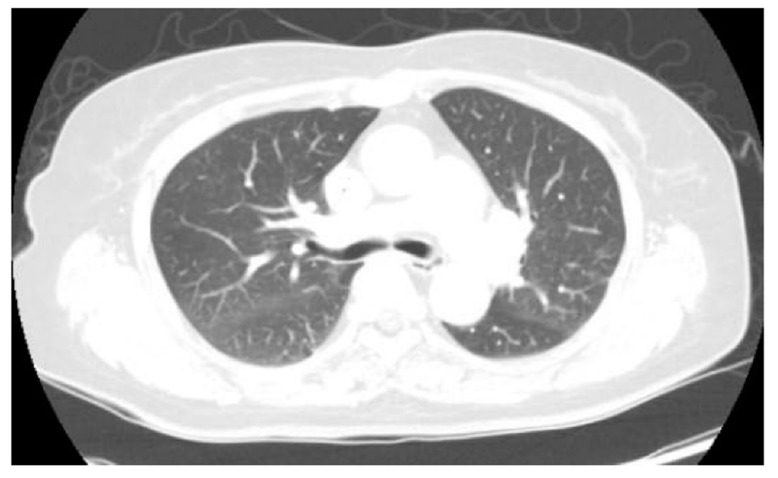
Follow-up chest CT after intravenous antibiotic treatment. A representative axial image reveals complete resolution of the lung lesions that were observed in the earlier PET/CT scan. **Discussion** Although ^18^F-FDG PET/CT has achieved great success in investigating malignant disorders, the imaging modality is not only specific for cancer diagnosis [13]. Since the activated inflammatory cells showed an increased expression and up-regulation of glucose transport receptors, several studies have reported the presence of high FDG uptake in acute and chronic infectious diseases such as mycobacterial, fungal, bacterial infection, sarcoidosis, radiation pneumonitis, and postoperative inflammation [14,15,16]. In this case report, we evaluated the utility of ^18^F-FDG PET/CT in a patient with a suspected metastatic infectious disease, and showed that it can visualize the correct foci leading to therapeutic management. CRBSI is associated with significant morbidity due to systemic infection and causes septic pulmonary emboli, which originate from the extrapulmonary site transported to the lung [17]. Like this case, clinical symptoms of septic pulmonary emboli are usually non-specific, and an active extrapulmonary focus of the infection might be apparent at the time of presentation, especially in cancer patients on chemotherapy via an indwelling central venous catheter for long durations. In conclusion, ^18^F-FDG PET/CT can detect septic pulmonary emboli in patients with catheter-related *Staphylococcus aureus* bacteremia. This case report suggests that cancer patients with CRBSI might benefit from ^18^F-FDG PET/CT for a timely evaluation of metastatic infection and optimal management. In accordance with previous studies suggesting the clinical value of ^18^F-FDG PET/CT in patients with Gram-positive bacteremia [5,10,11,18,19,20], we believe that the benefit from ^18^F-FDG PET/CT might be mediated by infective foci detection, earlier interventions to control infection, and the prolongation of antimicrobial treatment.

## Data Availability

The data that support the findings of this study are available from the corresponding author J.Y., upon reasonable request.

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
