# Peer review of "Septic Pulmonary Emboli Detected by 18F-FDG PET/CT in a Patient with Central Venous Catheter-Related Staphylococcus aureus Bacteremia"

_diagnostics, 2022, doi:10.3390/diagnostics12102479_

Round 1

Reviewer 1 Report

Dear Authors,

I read your article carefully.

It is interesting and I think it can be published.

I recommend:

- the detailed description of the figures (the captions are too short), and it is difficult to understand what you want to convey from the figures

- please correct the English mistakes

Author Response

Responses to Reviewer #1

Dear Authors,

I read your article carefully.

It is interesting and I think it can be published.

I recommend:

- the detailed description of the figures (the captions are too short), and it is difficult to understand what you want to convey from the figures

à We understand your concern. We added a more detailed descriptions of figure 2 and 3.

- please correct the English mistakes

à As you commented, we have corrected the English mistakes thoroughly.

Thank you for helpful comments. 

Reviewer 2 Report

The article entitled “Septic Pulmonary Emboli Detected by 18F-FDG PET/CT in a Patient with Central Venous Catheter-Related Staphylococcus Aureus Bacteremia” has been reviewed. In the present study, a case of 18F-FDG PET/CT detecting septic pulmonary emboli in a patient with Staphylococcus aureus catheter-related bloodstream infection (CRBSI). The patient, who had an implantable venous access port for chemotherapy, underwent 18F-FDG PET/CT to diagnose unsuspected infectious foci. The PET/CT examination made it possible to offer a suggestive diagnosis and yielded metastatic infectious foci.

The authors wrote this article with appropriate references and presented good-quality images.   

However, the present study lacks novelty, it was well established below cited references.

Ghanem-Zoubi N, Kagna O, Abu-Elhija J, Mustafa-Hellou M, Qasum M, Keidar Z, et al. Integration of FDG-PET/CT in the diagnostic workup for Staphylococcus aureus bacteremia: a prospective interventional matched-cohort study. Clin Infect Dis 2021;73:e3859-e3866

Berrevoets MAH, Kouijzer IJE, Aarntzen E, Janssen M, De Geus-Oei LF, Wertheim H, et al. (18)F-FDG PET/CT optimizes treatment in Staphylococcus aureus bacteremia and is associated with reduced mortality. J Nucl Med 2017;58:1504-1510.

Even though the authors applied the reported procedure to save patients suffering.

Author Response

Responses to Reviewer #2

The article entitled “Septic Pulmonary Emboli Detected by 18F-FDG PET/CT in a Patient with Central Venous Catheter-Related Staphylococcus Aureus Bacteremia” has been reviewed. In the present study, a case of 18F-FDG PET/CT detecting septic pulmonary emboli in a patient with Staphylococcus aureus catheter-related bloodstream infection (CRBSI). The patient, who had an implantable venous access port for chemotherapy, underwent 18F-FDG PET/CT to diagnose unsuspected infectious foci. The PET/CT examination made it possible to offer a suggestive diagnosis and yielded metastatic infectious foci.

The authors wrote this article with appropriate references and presented good-quality images.  

  • Thank you for kind comments.

However, the present study lacks novelty, it was well established below cited references.

Ghanem-Zoubi N, Kagna O, Abu-Elhija J, Mustafa-Hellou M, Qasum M, Keidar Z, et al. Integration of FDG-PET/CT in the diagnostic workup for Staphylococcus aureus bacteremia: a prospective interventional matched-cohort study. Clin Infect Dis 2021;73:e3859-e3866

Berrevoets MAH, Kouijzer IJE, Aarntzen E, Janssen M, De Geus-Oei LF, Wertheim H, et al. (18)F-FDG PET/CT optimizes treatment in Staphylococcus aureus bacteremia and is associated with reduced mortality. J Nucl Med 2017;58:1504-1510.

Even though the authors applied the reported procedure to save patients suffering.

  • As you said, it is clear that studies using 18F-FDG PET/CT to diagnose or treat catheter-related bloodstream infection have been published quite a bit. However, this is the first case in our hospital and the first case I experienced while attending the medical institution. So it was very interesting, and I submitted this report hoping that more readers of journal would share this case shortly.

Reviewer 3 Report

Even if the case is well-presented, I cannot think it could add anything to the knowledge of the readers.

The possibility of septic embolism in cancer patients with central venous catheter is not rare.

"However, the chest X-ray showed scattered nodular opacities in the left lung upper lobe and right lung lower lobe, suspicious of metastasis or eosinophilic lung disease" Why suspicious for eosinophilic?

The right lesion of chest X-Ray shows sigh of excavation, suspected for septic embolism. A CT should confirm this.

PET shows abnormal uptake not only in case of tumors, but also in inflammatory conditions, I cannot think the indication for the detection of septic pulmonary emboli can be supported by a single case report. 

Unenhanced chest CT would had been enough to establish the diagnosis 

Author Response

Responses to Reviewer #3

Even if the case is well-presented, I cannot think it could add anything to the knowledge of the readers.

The possibility of septic embolism in cancer patients with central venous catheter is not rare.

  • As you said, it is clear that studies using 18F-FDG PET/CT to diagnose or treat catheter-related bloodstream infection have been published quite a bit. However, this is the first case in our hospital and the first case I experienced while attending the medical institution. So it was very interesting, and I submitted this report hoping that more readers of journal would share this case shortly.

"However, the chest X-ray showed scattered nodular opacities in the left lung upper lobe and right lung lower lobe, suspicious of metastasis or eosinophilic lung disease" Why suspicious for eosinophilic?

  • I understand your concern. Considering your comment, I reviewed the X-ray in detail again. The possibility of eosinophilic lung disease seems low. Although eosinophilic lung disease has included in the differential diagnosis by the radiologist, the liklihood of eosinophilic lung disease appears low when both clinical information and laboratory test results are carefully considered. Therfore, any possibility of eosinophilic lung disease has deleted from the manuscript.

The right lesion of chest X-Ray shows sigh of excavation, suspected for septic embolism. A CT should confirm this.

  • As you said, I agree that it is the appropriate plan to perform the CT examination after X-ray. However, this patient had a history of ovarina cancer, so it was considered that PET/CT scan was performed for cancer work-up.

PET shows abnormal uptake not only in case of tumors, but also in inflammatory conditions, I cannot think the indication for the detection of septic pulmonary emboli can be supported by a single case report.

Unenhanced chest CT would had been enough to establish the diagnosis

  • I fully understand what you are pointing out. I also think there is insufficient evidence to suggest the indication for detecting septic pulmonary embolism with only a single case report. However, in this case, please consider that PET/CT scan performed to evaluate the infectious focus, which was unclear through physical examination. I agree that an unenhanced chest CT had been enough to establish the clinical diagnosis.

Thank you for helpful comments.

Round 2

Reviewer 3 Report

I thank the authors for their response, but I do not think they performed the suggested modifications.

The conclusions are still the same, and a case report is not sufficient to suggest the use of a PET-CT to detect infectious embolic foci. Again, the use of a chest CT should be enough. ". This case report suggests that cancer patients with CRBSI might benefit from 18F-FDG PET/CT for timely evaluation of metastatic infection and optimal management." this conclusion is too strong and not supported by any evidence.

You can simply describe the case, stating that in this specific case, the PET-CT highlighted embolic septic foci, but without any generalization. Moreover, the only X-Ray was suggestive, therefore, a clear indication for the PET-CT execution was missing. The staging could be performed with a contrast-enhenced whole body CT.

I cannot get why in a cancer patient the esosinophilic lung diseas ewas taken into consideration as a possible differential, as many other (more common, especially infective) conditions manifest with multiple lung opacities.

I underline again that the opacity in the right lung cleary shows a central area of escavation

Author Response

Responses to Reviewer #3

I thank the authors for their response, but I do not think they performed the suggested modifications.

The conclusions are still the same, and a case report is not sufficient to suggest the use of a PET-CT to detect infectious embolic foci. Again, the use of a chest CT should be enough. ". This case report suggests that cancer patients with CRBSI might benefit from 18F-FDG PET/CT for timely evaluation of metastatic infection and optimal management." this conclusion is too strong and not supported by any evidence.

  • I understand your concern. Of course, it is by no means emphasizing the need for PET/CT with just a single case report. Considering the results of several previous studies, I would like to suggest the benefit of PET/CT in patients suspicious of metastatic infection. In conclusion, the previous studies are added to support my thought.

You can simply describe the case, stating that in this specific case, the PET-CT highlighted embolic septic foci, but without any generalization. Moreover, the only X-Ray was suggestive, therefore, a clear indication for the PET-CT execution was missing. The staging could be performed with a contrast-enhenced whole body CT.

  • As you said, the PET/CT scan may seem lacking in generalization and may not apply to clinical indications in patients with catheter-related bloodstream infection. We are very well aware of this point. In our hospital, however, the contrast-enhanced whole body CT examination cannot be performed due to the patient’s medical insurance problem and lack of whole body CT equipment.

I cannot get why in a cancer patient the esosinophilic lung diseas ewas taken into consideration as a possible differential, as many other (more common, especially infective) conditions manifest with multiple lung opacities.

  • As I said the prior reviewer’s response, I explained that the primary radiologist seemed to have misinterpreted it at that time. Therefore, this possibility is very low, and I have replied that I have removed it from the manuscript. When I checked the first revised manuscript again, there was no correction for this point. I am very sorry for bothering you with mistake, and I have uploaded a newly revised manuscript, so please check it again.

I underline again that the opacity in the right lung cleary shows a central area of escavation

  • I have added your emphasis to the figure legend.

Thank you for helpful comments agaion.

Round 3

Reviewer 3 Report

I thank the authors for performing the required revisions